# Controlling the Thermal Stability of a Bainitic Structure by Alloy Design and Isothermal Heat Treatment

**DOI:** 10.3390/ma16082963

**Published:** 2023-04-07

**Authors:** Aleksandra Królicka, Francisca Garcia Caballero, Władysław Zalecki, Roman Kuziak, Radosław Rozmus

**Affiliations:** 1Department of Metal Forming, Welding and Technology, Faculty of Mechanical Engineering, Wroclaw University of Science and Technology, 50-371 Wroclaw, Poland; 2Department of Physical Metallurgy, National Center for Metallurgical Research (CENIM-CSIC), 28040 Madrid, Spain; 3Łukasiewicz Research Network—Upper Silesian Institute of Technology, 44-121 Gliwice, Poland

**Keywords:** medium-carbon steel, nanocrystalline bainite, CCT, TTT, dilatometry, bainite, tempering, design of materials

## Abstract

The aim of this work was to develop a novel bainitic steel that will be specifically dedicated to achieving a high degree of refinement (nano- or submicron scale) along with increased thermal stability of the structure at elevated temperatures. The material was characterized by improved in-use properties, expressed as the thermal stability of the structure, compared to nanocrystalline bainitic steels with a limited fraction of carbide precipitations. Assumed criteria for the expected low martensite start temperature, bainitic hardenability level, and thermal stability are specified. The steel design process and complete characteristics of the novel steel including continuous cooling transformation and time–temperature–transformation diagrams based on dilatometry are presented. Moreover, the influence of bainite transformation temperature on the degree of structure refinement and dimensions of austenite blocks was also determined. It was assessed whether, in medium-carbon steels, it is possible to achieve a nanoscale bainitic structure. Finally, the effectiveness of the applied strategy for enhancing thermal stability at elevated temperatures was analyzed.

## 1. Introduction

Nanocrystalline bainitic steels continue to attract substantial attention from scientists due to their excellent combination of strength properties and ductility [1]. A highly refined structure is generally achieved by a low-temperature transformation in the range of 125–350 °C and simultaneously through careful alloy design of high-carbon and high-silicon steels [2]. Typical nanocrystalline high-carbon bainitic steels are characterized by poor overall in-use properties including weldability [3], corrosion resistance [4,5], and limited resistance to tempering processes [6], which are one of the major causes for their restricted industrialization. The concept of enhancing in-use properties by improving the thermal stability of bainitic steels has been extensively described in previous work [7]. In the beginning, it should be mentioned that the development of the new chemical composition should also consider the fundamental requirements for the control of the kinetics of the bainite transformation. This is related to both the transformation temperature, transformation time, and bainite hardenability. Then, the chemical composition is adjusted to develop a thermally stable bainitic structure (Figure 1), as follows:Strength–ductility combination related to the structural morphology consisting of bainitic ferrite and retained austenite with a low fraction of carbides (carbide-free bainite);Improvement of the thermal stability of bainitic ferrite including carbon supersaturation, thickness refinement by controlling the martensite start (M_s_) temperature, and increasing the dislocations density [8];Improvement of the thermal stability of retained austenite including the control of its content, size, morphology, and chemical composition [9,10,11];The potential of secondary hardening [12].

Noticeably, the fundamental assumptions of the simultaneous reduction in M_s_ temperature, bainitic ferrite lath thickness, and carbon content are challenging. The temperature M_s_ depends primarily on the carbon content and, to a lesser extent, on alloying additives. Moreover, Morales-Rivas [13] pointed out that a reduction in the C content increases the industrial potential of nanocrystalline bainitic steels. In terms of thermal stability features, it is known that the nanoscale thickness of the bainitic ferrite laths inhibits the growth of carbides during decomposition under the influence of elevated temperature [14]. Several strategies for achieving nanocrystalline bainite in medium-carbon steels have also been conveyed in the literature. Zhang et al. [15] revealed that the austempering process of deformation-strengthened supercooled austenite leads to the reduction in the martensite start temperature and, at the same time, enables the transformation at a lower temperature to achieve nanocrystalline bainite. Another concept for obtaining nanoscale bainitic ferrite laths and austenite films is the use of isothermal heat treatment below the M_s_ temperature [16]. In fact, several published results presenting the thickness measurements using stereographic correction suggest that these steels are characterized rather by the sub-micrometer range [15,17,18]. In other research [19], the ausrolling process was also applied, where the thickness of the bainitic ferrite laths was at the boundary between the nano- and microscales. However, Peet et al. [20] applied extremely low bainitic transformation temperatures (even below 200 °C), where the degree of refinement was also at the boundary of both scales and was significantly thicker compared to high-carbon steels. It was stated that the refinement level of the bainitic ferrite laths depends not only on the bainite transformation temperature but also on the strength of the austenite [21].

The overriding purpose of the work is to develop a new steel grade that will be characterized by highly refined bainitic ferrite laths, film-like retained austenite, and fine blocky retained austenite. The fraction of carbide precipitations should also be significantly reduced (expected carbide-free bainite). Additionally, this concept also considered enhancing the thermal stability of the structure and improving weldability by ensuring the high bainitic hardenability required for the regeneration techniques [22,23,24]. Moreover, it is also significant, from the industrial point of view, in terms of producing parts with larger cross-sections [25]. As part of the current work, a complete transformation study of the designed and manufactured bainitic steel grade including experimental TTT (time–temperature–transformation) and CCT (continuous cooling transformation) diagrams was presented. Effort was focused on the determination of the relationship between bainite transformation temperature and structure evolution, considering the thickness of bainitic ferrite laths and film-like austenite as well as the coarseness of retained blocky austenite. Finally, the thermal stability during tempering processes was also evaluated.

### Design Process of Experimental Steel

The general assumption of the designed steel followed the requirements for nanocrystalline bainitic steels, excluding the carbon content [2]. The carbon content was assumed to be in the range of 0.4–0.6 wt.%, which was then adjusted based on the phase transformation simulations concerning the expected M_s_ temperature. To improve the thermal stability of the retained austenite [26,27] and to prevent the precipitation of cementite from austenite, it was assumed that the silicon content would be in the range of 1.5 to 3.0 wt.%. Although Mn is a stabilizer of austenite, it was proposed to reduce its content (0.5 wt.%) and increase the content of Cr (1.5 wt.%) to accelerate bainite transformation and achieve sufficient hardenability. It was stated that V (0.2 wt.%) has a beneficial effect on retarding the decomposition process during tempering [28,29]. Another aspect is also controlling the grain size of the prior austenite, which is significant in terms of steel’s susceptibility to grain coarsening. Moreover, additions of Cr and Mo may contribute to the sequence of carbide precipitation processes and thus cause secondary hardening during the tempering process [30]. In spite of their effect on the acceleration of bainitic transformation, Ni and Co were not considered [31,32], due to reducing the cost of experimental steel, similar to work [25].

Bearing in mind the above considerations, the Fe-(0.4–0.6)C-(1.5–3.0)Si-0.5Mn-1.5Cr-0.5Mo-0.2V alloy system was proposed.

The thickness of the bainitic ferrite plates depends indirectly on the bainitic transformation temperature. For this reason, it was assumed that the experimental steel should be characterized by a M_s_ temperature below 300 °C. There are numerous relationships between chemical composition and the temperature of martensitic transformation. To estimate the expected M_s_ temperature, a series of calculations were performed using known literature relationships (among others [33,34]) and the results of JMatPro^®^ (version 8.0, Sente Software Ltd., Guildford, UK) and Thermocalc (TCFE11 database, Solna, Sweden). The obtained results are presented in boxplots considering the median of the obtained results (Figure 2). It was noted that silicon does not significantly affect the M_s_ temperature and is not an important element in this case. However, in terms of a steel carbon content of 0.45 wt.%, there is a certain probability of reaching too high a M_s_ temperature (above 300 °C). For this reason, carbon contents lower than 0.50 wt.% were excluded for further consideration.

Theoretically, achieving carbide-free bainite requires approximately 2.0 wt.% of Si, which retards the precipitation of cementite from austenite [1]. However, this is also the reason for the reduction in the growth rate of bainite transformation [35]. From the point of view of industrialization, the transformation time should be as short as possible. On the other hand, the reduction in carbide precipitates and the stabilization of the retained austenite are crucial in terms of targeted mechanical properties (strength and ductility). For this reason, it was assumed that the silicon content would not be lower than 1.5 wt.%. The influence of silicon on the kinetics of phase transformations on the predicted TTT graphs was analyzed (Figure 3a–c). The expected bainitic hardenability ought to be characterized by shifting the pearlite transformation toward the longest times (t_diff_), and the bainite transformation toward the shortest times (t_shear_). Hasan et al. [36] calculated that Si generally reduces bainitic hardenability, while Mo up to 0.5 wt.% improves bainitic hardenability. The red arrow in Figure 3a–c indicates the differences between the position of both analyzed transformations. It was calculated that silicon lowers the bainitic hardenability, and at 2.5 wt.% of Si, the pearlite transformation start time is generally comparable to the bainite transformation start time. Based on these calculations, it was decided to assume 2.0 wt.% of Si, which seems to be a reasonable compromise between the bainitic hardenability and promotion of the formation of carbide-free bainite.

To determine the suitability of designed steel for the potential welding process with the regeneration technique, particularly the ability to produce longer welds, and to enable the heat treatment of larger cross-sectional parts, the critical cooling rate was crucial. Figure 3d,e show the predicted CCT diagrams for two different carbon contents (0.5 and 0.55 wt.%). The average cooling rate for slow cooling in the air is indicated in the figures by the yellow line. In the case of 0.5C steel, a structure consisting of martensite and a low fraction of bainite is expected in this area, while 0.55C steel already contains diffusion transformation products (pearlite). Therefore, the selected carbon content of the experimental steel was 0.5 wt.%.

It was also expected that the austenitization process should include homogeneous austenite with dissolved carbides. The designed steel contains strong carbide-forming alloying additives; therefore, analysis of the carbide fraction in the temperature function in the state of thermodynamic equilibrium was performed (Figure 4). Vanadium is prone to formation of the (M,V)C carbides; however, calculations did not indicate this carbide as stable, which may be explained by the effect of Cr [37]. The calculations reveal that the most stable carbide at elevated temperature is M_6_C, whose dissolution temperature depends on the Mo content. For the assumed Mo content (0.5 wt.%), theoretically, the austenitization temperature should not be lower than 941 °C, to effectively dissolve all carbides.

Finally, the designed chemical composition was proposed to be Fe-0.5C-2.0Si-1.5Cr-0.5Mn-0.5Mo-0.2V.

## 2. Materials and Methods

Experimental steel was produced as a 28 kg cast ingot subjected to homogenization at 1200 °C for 1 h and then hot-forged to a final cross-section (diameter of approx. 42 mm). The experimental melting process was conducted at Łukasiewicz Upper Silesian Institute of Technology (Gliwice, Poland). Table 1 presents a comparison of the designed chemical composition with the measured chemical composition of the laboratory cast. The microstructure after homogenization and hot forming consisted of fine-dispersive pearlite (Figure 5) with a hardness of 320 ± 5 HV10. The formation of proeutectoid ferrite was not evident.

### Methods

The dilatometric study of the steel in the as-delivered state was conducted using a DIL 805A/D/T dilatometer (TA Instruments, New Castle, U.S.). Tubular profile samples (outer diameter 4 mm, inner diameter 2 mm, length 10 mm) were used. The determination of CCT diagrams contained an austenitization process at 950 °C for 1200 s and various cooling rates (481 °C/s, 190 °C/s, 100 °C/s, 50 °C/s, 20 °C/s, 10 °C/s, 5 °C/s, 2 °C/s, 1 °C/s, 0.5 °C/s, 0.25 °C/s, 0.1 °C/s, 0.05 °C/s). All experiments were performed in vacuum. For cooling rates higher or equal to 1 °C/s, gas cooling was applied. The temperature M_s_ was determined as the average of the cooling rates covering the range from 481 °C/s to 2 °C/s. The TTT diagram was developed for isotherms with a step of 25 °C. The lowest isothermal annealing temperature was 275 °C and the highest was 800 °C. In terms of the TTT diagram, the austenitization conditions were similar to those for the CCT diagram (950 °C for 1200 s). Then, the CCT and TTT diagrams were developed based on the analysis of dilatometric curves, hardness measurements, and microstructure observations in qualitative and quantitative approaches.

Metallographic samples intended for LOM (Light optical microscopy) and SEM (Scanning electron microscopy) investigations were prepared by grinding and polishing with diamond paste (3 μm and 1 μm). Subsequently, the samples were etched with 3% HNO_3_ (nital). Microstructural observations using SEM were performed using the topographic mode (secondary electron detector) at an accelerating voltage of 10–15 kV and a working distance of 8–10 mm. The investigations were carried out using high-resolution scanning electron microscopes JSM-7200F (JOEL, Tokyo, Japan) and Hitachi S-4800 (Hitachi High-Tech, Tokyo, Japan).

Estimation of the thermal stability of the designed steel including samples with dimensions of 1 cm × 1 cm × 1 cm was carried out using laboratory furnaces. The tempering process was performed for the sample austenitized at 950 °C for 30 min and isothermally treated at ~280 °C based on performed microstructural analysis focused on improving thermal stability. The tempering range was 350–650 °C and its time was 1 h. The hardness measurements of the samples subjected to the tempering process were conducted using the Vickers scale according to the EN ISO 6507-1:2018 standard, using a Matsuzawa MMT-X7B hardness tester (Matsuzawa, Akita, Japan). A load of 9.81 N (HV1) was applied for 15 s.

## 3. Results and Discussion

### 3.1. Examination of Phase Transformation Kinetics Based on Dilatometry

Table 2 compares the predictions of A_c1_ (critical temperature at which ferrite is still stable), A_c3_ (temperature at which ferrite is completely transformed into austenite), and M_s_ temperatures with experimental results. The estimated temperature M_s_ is the median of the series of calculations indicated in Figure 2. The A_c1_ temperature was predicted using Program MAP_STEEL_AC1TEMP [38] based on the experimental chemical composition, and the A_c3_ temperature using Subroutine MAP_STEEL_AC3 [39]. The heating rate was similar to the dilatometry tests (2.5 °C/s). In general, it was assessed that the predictions were generally close to the actual values. For the M_s_ temperature, the estimations may be considered accurate. The above results confirm that the assumption of reaching a M_s_ below 300 °C was fulfilled. The A_c3_ temperature indicates that austenitization at the temperature of 950 °C led to obtaining homogeneous austenite without carbide precipitations (carbide dissolution temperature is 941 °C according to Figure 4).

The critical cooling rate was determined based on the experimental CCT diagram (Figure 6a). This diagram was developed based on the analysis of dilatometry curves and the observation of microstructures using LOM and SEM. Crucial microphotographs for selected thermal cycles are presented in Figure 7. For the fastest cooling rate (~480 °C/s) to 2 °C/s, only the martensite transformation was identified. The presence of retained austenite was not excluded but was not considered in this analysis. The temperature M_s_ was constant (263 ± 2 °C) up to 2 °C/s and then gradually decreased (approx. 200 °C for a cooling rate of 0.25 °C/s). In terms of the bainitic hardenability, the cooling rate to obtain a bainitic structure was lower than 2 °C/s and higher than 0.5 °C/s. However, the bainite transformation was not completed after continuous cooling at any investigated rate. This suggests that the designed steel requires isothermal heat treatment to achieve a fully bainitic structure. Nevertheless, the obtained diagram proves that the steel had a high overall hardenability. In the case of the welding process with the regeneration technique, the time was also sufficient to produce larger welded parts and a direct cooling to the bainite transformation temperature. It should also be mentioned that the obtained results differed from the phase transformation calculations (Figure 3). Additionally, in the work [40], it was found that the type of software used also has a significant impact on the results of multi-phase steels.

The assessment of the microstructures after continuous cooling cycles was also performed with a quantitative approach (Table 3). Retained austenite was not included in this analysis. This procedure contained the qualitative identification of structures based on SEM observations and analysis of dilatometry curves. Then, graphical image editing was used by the ImageJ software (version 1.54b, National Institutes of Health, University of Wisconsin, Madison, USA). The obtained results were based on representative micrographs, and the result was about 1% due to the low accuracy of this analysis. For a cooling rate of 0.25 (°C/s), the total content of bainite and martensite was determined. In addition, these test results confirm the high hardenability of the designed steel. On the other hand, the bainitic hardenability was insufficient to achieve a fully bainitic structure after continuous cooling.

A complete TTT diagram with a measurement increment of 25 °C was developed (Figure 6b). The shortest time to start diffusional transformations was in the range of 45–60 s (at 700 °C), while the shortest start time to bainite transformation was approx. 80 s (at 350 °C). This proves a moderate bainitic hardenability, which does not coincide with the previous phase simulations. In terms of the isothermal heat treatment in the range of 650–800 °C (Figure 8a), pearlite was identified, and the distance between the lamellae of cementite increased with the holding temperature. At lower temperatures, pearlite became finer, and a spheroidal morphology of cementite was also identified (Figure 8b). It should also be highlighted that no phase transformation occurred in the range of 450–550 °C (Figure 8c). However, at a temperature of 400 °C, only some bainite was formed (Figure 8d). Below a temperature of 400 °C, up to a temperature close to M_s_ (275 °C), a complete bainitic transformation occurred (Figure 8e,f). The bainitic structure after isothermal heat treatment below 375 °C consisted of bainitic ferrite and retained austenite with film-like and blocky morphology.

Then, effort was focused on the characterization of the bainitic transformation in qualitative and quantitative approaches. Table 4 includes temperatures related to the bainite transformation along with the identification of the minimum time to form bainite (t_Bs_), and the time to finish the transformation (t_Bf_). From 400 °C to 425 °C, the time for bainite transformation was longer than the assumed maximum time (65–234,000 s). In the case of a temperature of 450 °C, the time to start the transformation was as much as approx. 22 h and the micrograph does not clearly convey the bainitic structure (Figure 8c); hence, the fraction of this transformation was insignificant. Therefore, it was stated that there is no transformation in this temperature range, which is consistent with other studies [41]. This may be explained by the solute drag-like effect (SDLE) caused by alloying additives (Cr, Mo, Ni) that significantly hinder the diffusion of carbon [42]. As a result, the incomplete transformation phenomenon or even a lack of transformation occurring between the ferrite/pearlite and the bainite transformation fields may be observed [43]. Then, starting at 375 °C, the bainite transformation ended. The shortest t_Bs_ and t_Bf_ occurred at 350 °C. The bainite transformation ended at this temperature after about 51 min counting the isothermal holding process. Subsequently, at lower temperatures, the bainite transformation time increased. The longest time of bainitic transformation was determined at the temperature of 275 °C (approx. 7 h). Thus, the bainitic transformation time increased as the transformation temperature decreased [44]. It is well-known that the temperature of the bainite transformation affects the morphology of the structure and the degree of its refinement. For this reason, the next section focuses on a comparative analysis of the structures in relation to the transformation temperature.

### 3.2. Examination of Bainitic Ferrite Thickness and Morphology of Retained Austenite

The influence of the bainitic transformation temperature on the thickness of the bainitic ferrite laths was determined. Measurements of the thickness of the bainitic ferrite laths were carried out on dilatometry samples at the bainitic transformation temperature in the range of 275–375 °C. The presented measurements included at least 120 linear intercepts and then were subjected to a stereographic correction to the mean linear intercept $t=\frac{2\cdot \bar{\mathrm{Lt}}}{\pi }$, widely described in [18]. Moreover, the measurements were compared with the prediction. Calculations of the expected thickness of the bainitic ferrite laths were determined from the equation proposed by Yang [45]: t = f(T,σ_y_^γ^, ΔG^γ→α^) = 222 + 0.01242 × T + 0.01785 × ΔG^γ→α^ − 0.5323 × σ_y_^γ^
(1)
where σ_y_^γ^ indicates the austenite strength (MPa), ΔG^γ→α^ is the change in chemical free energy (J∙mol^−1^), and T is the isothermal transformation temperature (°C). The austenite strength was determined using Equation (2), described in [46]. ΔG^γ→α^ was calculated using mucg83 software [47].
σ_y_^γ^ = (67.8 + 493ω_N_ + 354ω_C_ + 20.8ω_Si_ +3.70ω_Cr_ + 14.5ω_Mo_ + 1.9ω_Mn_ + 0.2ω_Ni_ + 0.1ω_V_ + 0.4ω_Al_ + 0.4ω_Cu_ + 0.2ω_Ti_)(1 − 0.26 × 10^−5^T^2^ − 0.326 × 10^−8^ T^3^)(2)
where ω indicates the content of given alloying additives (in wt.%) and T is the isothermal transformation temperature.

Both calculations and micrograph observations confirmed a strong relationship between the transformation temperature and the thickness of the bainitic ferrite laths (Figure 9 and Figure 10). The calculations suggested that a temperature transformation below 300 °C was favorable to obtain the thickness of the bainitic ferrite laths at the nanoscale. The thickness of the bainitic ferrite laths of the samples subjected to isothermal heat treatment at 300 °C was at the boundary of the nano- and submicron scales. In terms of isothermal heat treatments above 300 °C, the thickness gradually increased.

Due to the scatter of measurements, the median was considered as a measure of the thickness of the bainitic ferrite laths. The measure of dispersion was the first and third quartile, as shown in Figure 10a. In the temperature range of 275–350 °C, a slight increase in the thickness of the bainitic ferrite laths was observed, while at the temperature of 375 °C, their significant coarsening was noticed. Also at the highest temperature, there were outliers and a wide range of whiskers. Merely the isothermal transformation at the lowest temperature (275 °C) enabled the obtaining of the bainitic ferrite at the nanoscale. However, it is visible that the third quartile and the upper whisker were also on the sub-micron scale. This suggests that despite the median (87 nm) and average value (89 ± 6), the structure was close to the borderline between both scales. Due to most of the measurements being smaller than 100 nm, this structure was classified as nanocrystalline bainite.

Comparisons of calculations and measurements were partly consistent (Figure 10b). In the case of the lowest isothermal heat treatment temperatures, an overestimation was indicated, while an underestimation was observed at the highest temperatures. Nevertheless, calculations of bainitic ferrite thickness at the step of developing the chemical composition effectively estimated the possibility of obtaining a nanocrystalline structure at the assumed transformation temperature.

The refinement degree of the structure also includes the morphology and dimensions of the retained austenite. Retained austenite plays a crucial role in controlling the thermal stability of the bainitic steels. While the thickness of the film-like retained austenite is smaller than the thickness of the bainitic ferrite and thus is on the nanoscale, the assessment of the austenite blocks requires more attention. It is known that the chemical composition (carbon content) together with dimensions affect the thermal [7] and mechanical stability [48] of the blocky austenite. For this reason, the distribution of the average area surface of blocky austenite developed based on the graphical image editing was carried out. Blocky austenite was manually traced and then its surface area was measured using ImageJ software; an example of this analysis is presented in Figure 11. The detection threshold was set at 0.2 µm^2^.

Surface area measurements of blocky austenite were characterized by a high dispersion. In all samples, the distributions were determined by right skewness. The areas highlighted as “coarse“ adjusted to each singlet heat treatment variant. Therefore, both the average surface area of the blocks and the average surface area of the coarse austenite also including the content of blocky retained austenite were analyzed (Figure 12). The median of the measurements of each variant was also presented for an unambiguous assessment. In the case of bainite transformation in the range of 275–325 °C, the median was comparable (approx. 0.5 µm^2^). In terms of the transformation temperatures of 350 °C and 375 °C, the median of the surface area of the blocky austenite exceeded 1 μm^2^. When analyzing the average area of coarse austenite, its lowest values were identified for transformation temperatures of 300 °C and 325 °C (approx. 1.5 µm^2^). At higher transformation temperatures, the average surface area of coarse blocks increased to approx. 6 µm^2^ at a temperature of 375 °C. Interestingly, at the lowest transformation temperature (275 °C), locally coarse austenite regions with a significantly larger surface area than adjacent temperatures (300–325 °C) were identified. These areas resulted from the uncompleted bainite transformation and a slight difference from the M_s_ temperature (263 °C, 12 °C difference). Overall, the blocky austenite content was dependent on the transformation temperature. The lowest content of blocks was observed for the transformation at 275 °C (16 ± 3%) and the highest at 375 °C (34 ± 3%) (Figure 12).

Analyzing both the degree of refinement and the morphology of austenite, designed steel isothermally treated at approx. 280 °C was subjected to further investigations of thermal stability during tempering processes. Despite the presence of coarse retained austenite, which was characterized by a slightly less favorable distribution, the prevailing feature to inhibit decomposition processes during tempering was the thickness of the bainitic ferrite laths (expected nanoscale). Moreover, at this temperature, the lowest content of blocky retained austenite was identified.

### 3.3. Tempering Process—Thermal Stability Evaluation

The isothermally treated experimental steel at 280 °C was subjected to a tempering process at various temperatures. The graph of hardness versus tempering temperature is shown in Figure 13, where the hardness before tempering is also indicated (isothermal heat treatment). There was a slight decrease in hardness in the first stages of the tempering of experimental steel. Up to a temperature of 500 °C, the hardness was in the range of 555÷569 HV1; in contrast, the hardness after the isothermal heat treatment (austempering) was 583 ± 5 HV1. This indicates the high tempering resistance of experimental steel and relatively insignificant structural changes in this regard. Subsequently, at 550 °C, there was a sudden increase in hardness (578 ± 5 HV1), which was generally equal to the hardness after austempering (also 583 ± 5 HV1). Then, at higher tempering temperatures, a hardness drop was revealed, and at 650 °C, the hardness was 418 ± 5 HV1. In contrast, many investigations indicate a continuous decrease in the hardness of bainitic steels [27,49,50,51].

Based on the microstructural approach (Figure 14), designed steel during the tempering processes was characterized by a comparable morphology of structure up to a temperature of 500 °C (not presented here) in relation to the austempering process (Figure 14a). The decomposition processes undergone in the low-temperature range were sensitive and significant differences could not be clearly determined using the SEM method (Figure 14b). At the tempering temperature of 400 °C, precipitation processes began to be identifiable in the bainitic ferrite laths area (Figure 14c). On the other hand, a further increase in the tempering temperature to 550 °C led to significant changes in the structural morphology (Figure 14d). The typical lath structure of alternately distributed bainitic ferrite and austenite films disappeared. Carbide precipitations are clearly visible both on the boundaries of the bainitic ferrite laths and inside them. In terms of higher tempering temperatures, the fraction and dimensions of carbide precipitations increased (Figure 14d).

A significant increase in hardness was identified at 550 °C. It should be emphasized that the following decomposition mechanisms were identified: (i) almost complete film-like austenite decomposition into cementite and ferrite; the growth of cementite was limited by the bainitic ferrite laths boundaries; (ii) carbide precipitation from carbon-supersaturated bainitic ferrite; (iii) coarse blocky austenite decomposition.

Overall, the mentioned decomposition mechanisms resulted in a decrease in hardness in relation to the material not subjected to tempering. However, it was observed that the hardness at 550 °C was comparable to the state before tempering. This may be explained by the strengthening effect of fine-dispersed carbide precipitations (related to alloying elements such as Mo and V—secondary hardening), similar to [29,52].

In summary, the designed material is characterized by the satisfactory thermal stability of the structure up to a temperature of 550 °C for 1 h. Up to this temperature range, there are subtle decomposition mechanisms, and both retained austenite and bainitic ferrite do not exhibit significant differences compared to the state before tempering. The hardness in this range is merely about 5% lower, which also confirms the high thermal stability of the structure. On the other hand, tempering at temperatures close to and higher than 550 °C already leads to serious decomposition processes, which are compensated by the secondary hardening effect. The assumed concept including refinement of the bainitic ferrite laths, stabilization of the retained austenite, and secondary hardening was accordingly implemented in the designed material.

## 4. Conclusions

Based on the developed results, the following conclusions were formulated:A novel medium-carbon bainitic steel (Fe-0.5C-2.0Si-0.5Mn-1.5Cr-0.5Mo-0.2V) was developed along with the comprehensive characteristics of the phase transformations. The process of designing the chemical composition included the reduction in the M_s_ temperature, bainitic hardenability, and the improvement of the thermal stability of the structure under the influence of elevated temperatures.Experimental TTT and CCT diagrams were developed based on dilatometry investigations. The bainite transformation time was determined at different temperatures, which, in terms of the lowest bainite transformation temperature, was longest (275 °C, ~7 h) when it was shortest at 350 °C (51 min). The critical cooling rate of steel was between 1 °C/s and 2 °C/s, while the cooling rate required to avoid diffusional transformations was between 1 °C/s to 0.5 °C/s. Despite the assumptions, the designed steel had a moderate bainitic hardenability, and the potential processes of continuous cooling toward a bainitic structure were excluded.Overall, the hardenability of designed steel (except for bainitic hardenability) was relatively high—air cooling between the austenitization temperature and the isothermal heat treatment was ensured. This suggests that this material is suitable for the heat treatment of parts with larger cross-sections. In addition, it is also a premise for potential welding processes with the regeneration technique and the possibility of performing longer welded joints.The refinement level of designed steels was evaluated as nano- and sub-micro scales. In terms of temperatures below 300 °C, the average thickness of bainitic ferrite was 89 ± 6 nm (median 87 nm). Starting at 300 °C, the thickness gradually increased up to 350 °C and may be referred to the sub-micro scale. Above 350 °C, a sudden coarsening of bainitic ferrite laths was observed (155 ± 10 nm). The dimensions of retained blocky austenite were inversely proportional to transformation temperature. At the temperature closest to M_s_ (275 °C), locally coarse blocky austenite was identified, which was characterized by a larger surface area in relation to the adjacent temperatures. The blocky retained austenite increased with the transformation temperature from 16 ± 3% to 34 ± 3%.The thermal stability of the structure at elevated temperatures was evaluated as satisfactory. The microstructure up to the tempering temperature of 550 °C did not contain severe differences concerning samples not subjected to tempering processes. After tempering at 400 °C, the first, subtle symptoms of the decomposition of retained austenite with film-like morphology were noticed. The hardness after tempering at 550 °C increased to a value comparable to the samples before tempering. This suggests that the alloying additives (Mo, V, and Cr) tended to secondary hardening at higher tempering temperatures.

## Figures and Tables

**Figure 1 materials-16-02963-f001:**
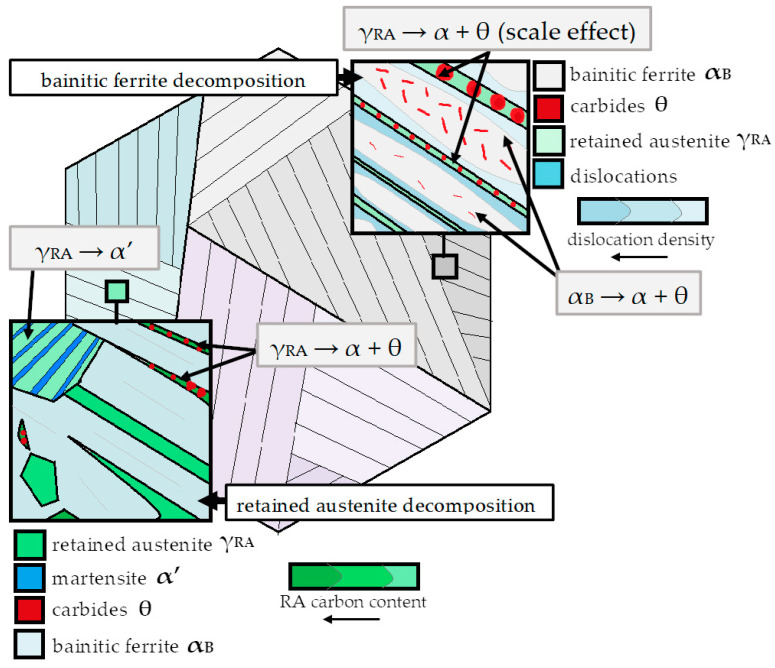
Scheme of main decomposition processes of retained austenite and bainitic ferrite in carbide-free bainitic structures.

**Figure 2 materials-16-02963-f002:**
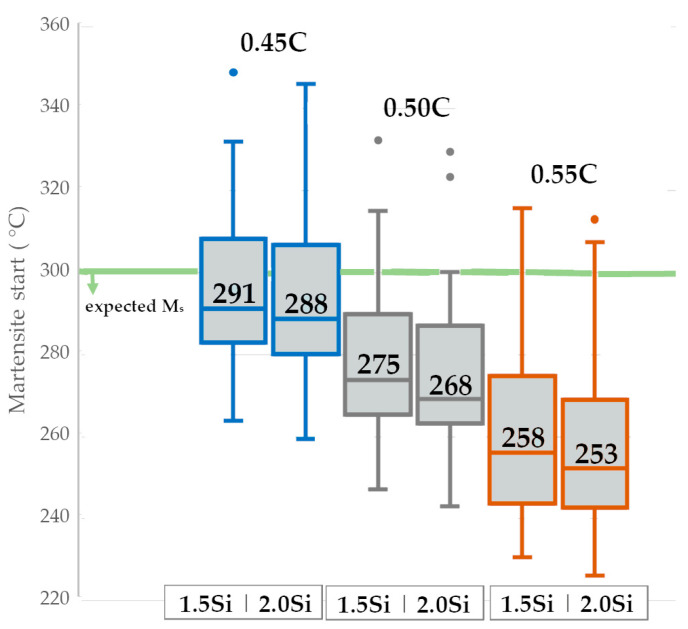
Boxplots of M_s_ temperature in relation to carbon (0.45÷0.55 wt.%) and silicon (1.5 and 2.0 wt.%) content.

**Figure 3 materials-16-02963-f003:**
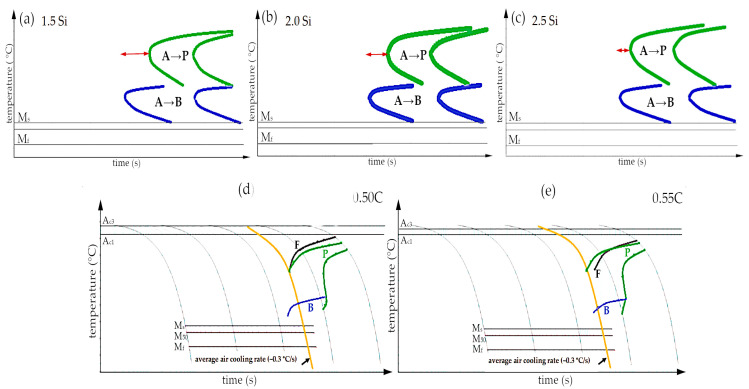
Calculations of phase transformation. The TTT diagrams of Fe-0.45C-XSi-0.5Mn-0.5Mo-0.2V considering: (**a**) 1.5 wt.% Si; (**b**) 2.0 wt.% Si; (**c**) 2.5 wt.% Si. The CCT diagrams of Fe-XC-2.0Si-0.5Mn-0.5Mo-0.2V steel considering: (**d**) 0.5 wt.% C and (**e**) 0.55 wt.% C. JMatPro, version 8.0.

**Figure 4 materials-16-02963-f004:**
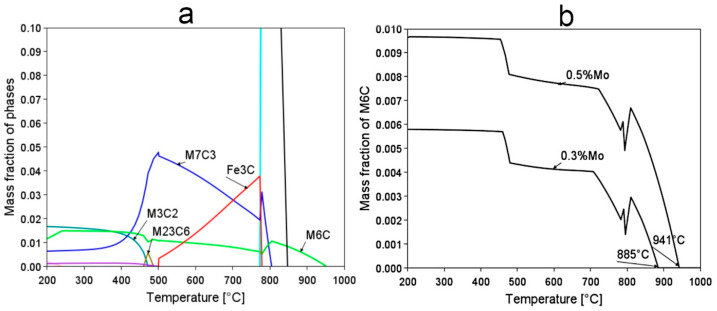
(**a**) Calculated fraction of carbide phases as a function of temperature of steel 0.5C-2.0Si-1.5Cr-0.5Mn-0.2V-0.45Al-0.5Mo. (**b**) Comparison of the M_6_C carbide dissolution temperature for the two Mo contents of 0.3 wt.% and 0.5 wt%. Thermocalc, database version TCFE11.

**Figure 5 materials-16-02963-f005:**
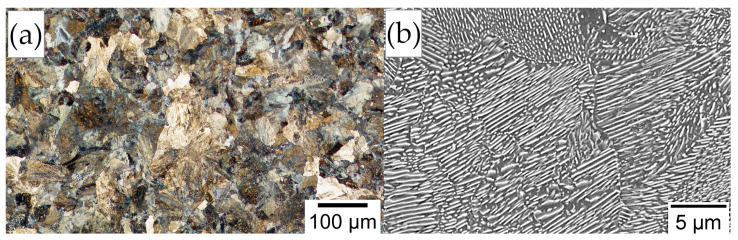
Microstructure of manufactured steel after homogenization and hot forging process. Fully fine-dispersed pearlitic structure. (**a**) Light optical microscopy. (**b**) Scanning electron microscopy.

**Figure 6 materials-16-02963-f006:**
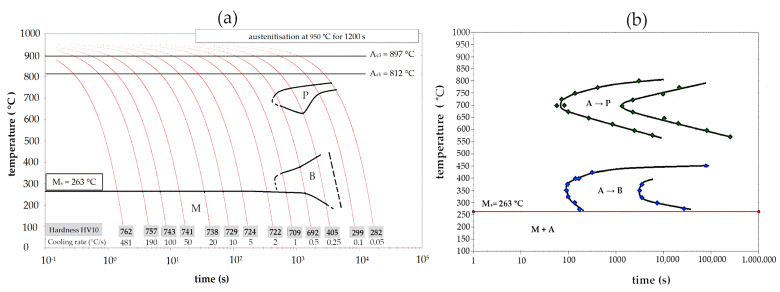
Experimental CCT (**a**) and TTT (**b**) diagrams based on dilatometry. Austenitization was carried out at 950 °C.

**Figure 7 materials-16-02963-f007:**
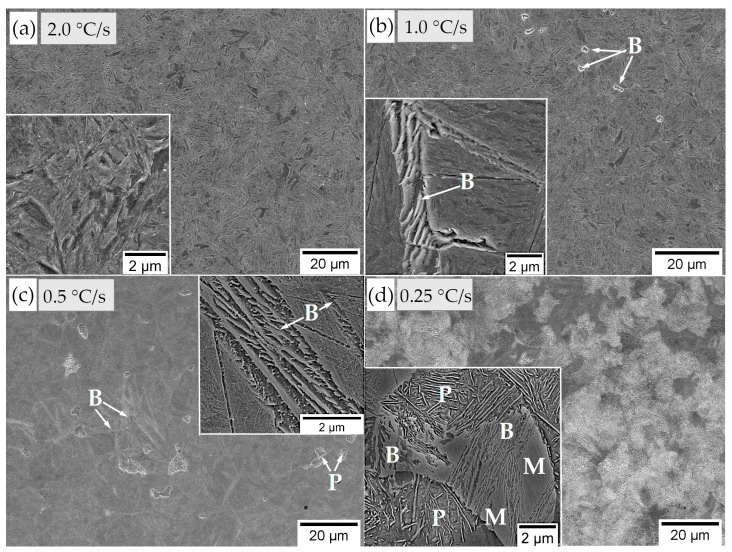
Microstructures of experimental steel after selected continuous cooling cycles. (**a**) 2.0 °C/s; martensite. (**b**) 1.0 °C/s; martensite and bainite. (**c**) 0.5 °C/s; martensite, bainite, and locally visible pearlite. (**d**) 0.25 °C/s; martensite, bainite, and pearlite. Scanning electron microscopy, etched by nital. B—bainite; P—pearlite; M—martensite.

**Figure 8 materials-16-02963-f008:**
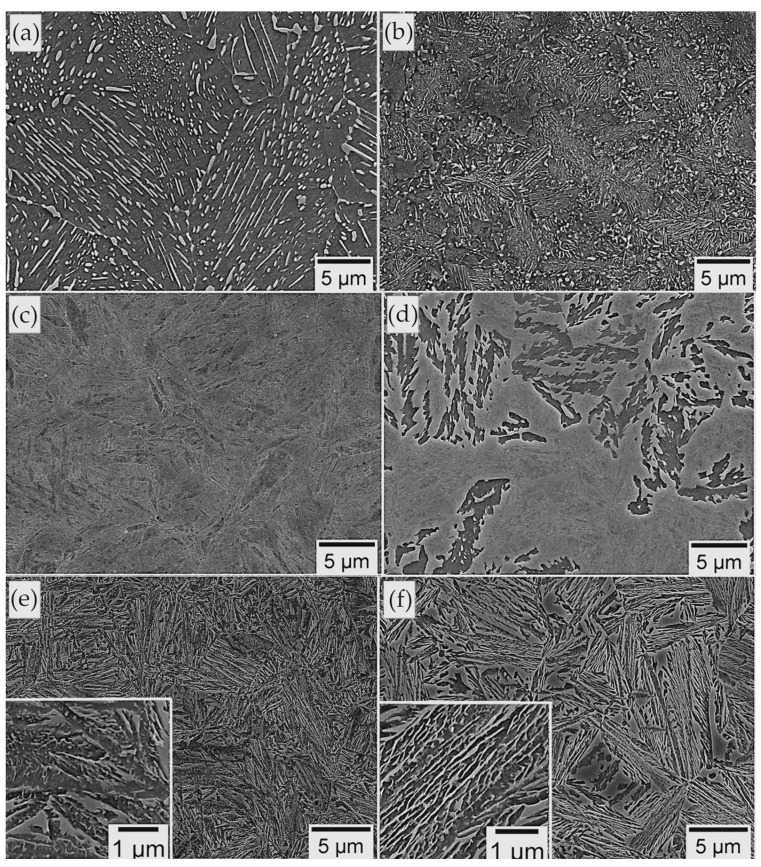
Microstructures of experimental steel after isothermal heat treatment at different temperatures. (**a**) 750 °C for 20 h; pearlite. (**b**) 575 °C; pearlite for 68 h, degenerated pearlite, partially spheroidal. (**c**) 450 °C for 68 h; lack of identified transformation, martensite. (**d**) 400 °C for 64 h; partially transformed upper bainite and martensite. (**e**) 350 °C for 3 h; lath morphology of bainitic ferrite and retained austenite (blocks and films). (**f**) 275 °C for 15 h; lath morphology of bainitic ferrite, retained austenite (blocks and films). Scanning electron microscopy, etched by nital.

**Figure 9 materials-16-02963-f009:**
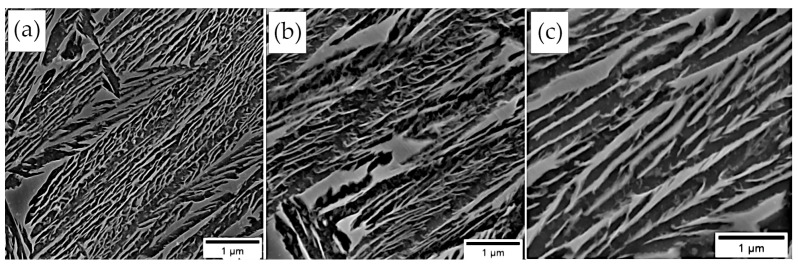
Micrographs of bainitic ferrite laths in different transformation temperatures: (**a**) 275 °C; (**b**) 325 °C; (**c**) 375 °C. Scanning electron microscopy, etched by nital.

**Figure 10 materials-16-02963-f010:**
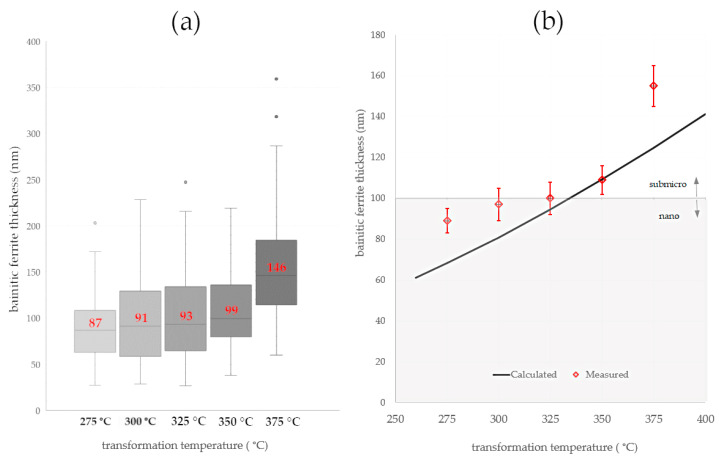
(**a**) Boxplots representing the median and interquartile range of bainitic ferrite thickness in relation to transformation temperature. (**b**) Comparison of calculated and measurement results expressed as average value and 95% confidence interval.

**Figure 11 materials-16-02963-f011:**
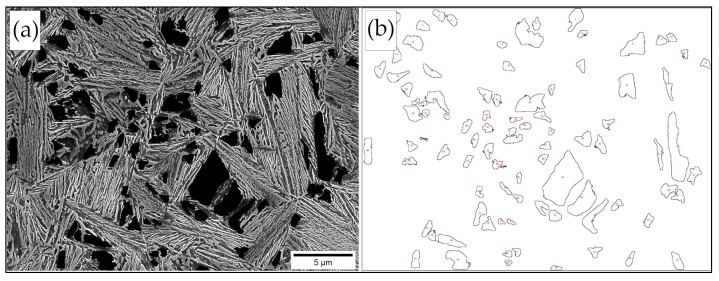
Example of quantitative analysis of retained blocky austenite dimensions. (**a**) Micrograph highlighting the retained blocky austenite; (**b**) outlines of retained blocky austenite subjected to measurements using ImageJ software. Sample after isothermal treatment at 275 °C.

**Figure 12 materials-16-02963-f012:**
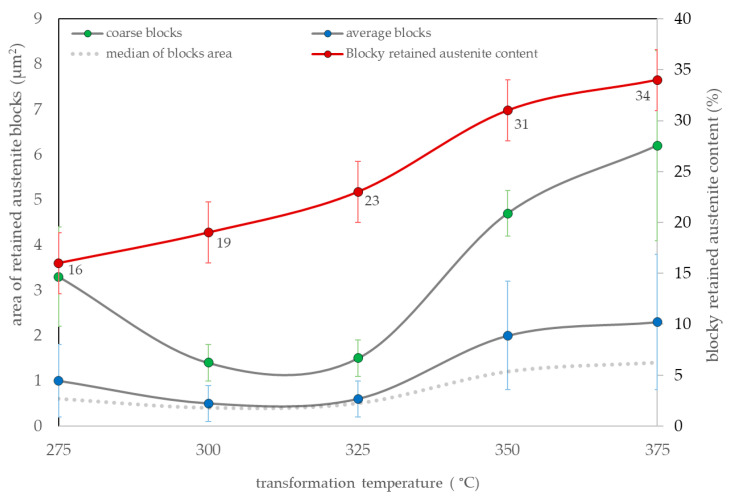
Quantitative analysis of retained austenite blocks expressed as surface area based on graphical image editing. Coarse blocks of retained austenite, the average surface area of blocky austenite, a median of measurements, and the total content of blocky retained austenite.

**Figure 13 materials-16-02963-f013:**
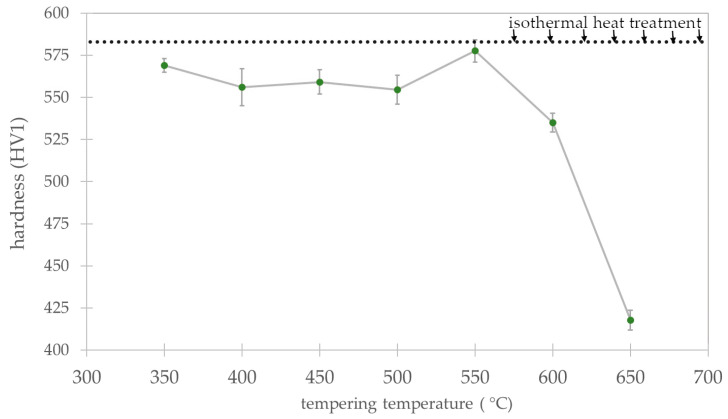
Hardness distribution of experimental steel after isothermal heat treatment at 280 °C subjected to the tempering process.

**Figure 14 materials-16-02963-f014:**
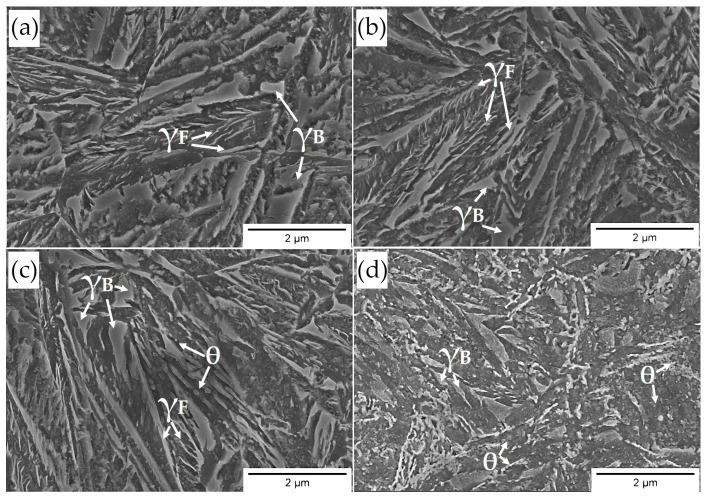
Microstructure of experimental steel subjected to isothermal heat treatment and tempering process: (**a**) Isothermal heat treatment at 280 °C before tempering; (**b**) tempering at 350 °C; (**c**) tempering at 400 °C; (**d**) tempering at 550 °C. SEM, secondary electrons detector. γ_F_—film-like retained austenite; γ_B_—blocky retained austenite; θ—carbides.

**Table 1 materials-16-02963-t001:** Chemical composition (wt.%) of developed steel.

	C	Mn	Si	Cr	Mo	V
Designed	0.50	0.50	2.00	1.50	0.50	0.20
Experimental	0.49	0.51	1.99	1.54	0.51	0.22

**Table 2 materials-16-02963-t002:** Determination of M_s_, A_c1,_ and A_c3_ temperatures.

	M_s_ (°C)	A_c3_ (°C)	A_c1_ (°C)
Predicted	268 ± 15	905 ± 35	788 ± 22
Dilatometry	263 ± 2	897 ± 2	812 ± 2

**Table 3 materials-16-02963-t003:** Quantitative data of performed thermal cycles (continuous cooling) considering phase fractions and hardness.

Cooling Rate (°C/s)	Structure (Graphical Editing of Images)	Hardness HV10
from ~480 to 2	mainly martensite	form 762 to 722 ± 11
1	martensite (96%) + bainite (4%)	709 ± 11
0.5	martensite (89%) + bainite (8%) + pearlite (3%)	692 ± 11
0.25	martensite and bainite (35%), pearlite (65%)	405 ± 3
0.1	mainly pearlite	299 ± 5
0.05	mainly pearlite	282 ± 4

**Table 4 materials-16-02963-t004:** Bainite transformation times at different temperatures during isothermal holding.

Transformation Temperature (°C)	Time to Start (t_Bs_)	Time to Finish (t_Bf_)
450	80,000 s|22 h	longer than 68 h
425	300 s|5 min	longer than 68 h
400	150 s|2.5 min	longer than 68 h
375	90 s|1.5 min	3455 s|57 min
350	80 s|1.3 min	3091 s|51 min
325	90 s|1.5 min	3308 s|55 min
300	140 s|2.3 min	7443 s|126 min
275	180 s|3 min	261,666 s|7 h 15 min

## Data Availability

Data are contained within the article.

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
