# Peer review of "Controlling the Thermal Stability of a Bainitic Structure by Alloy Design and Isothermal Heat Treatment"

_materials, 2023, doi:10.3390/ma16082963_

Round 1
Reviewer 1 Report
Pleas look at the attached pdf document

Author Response
Dear Reviewer,
Thank you for your review of our manuscript and a few important comments. We hope that with their help we were able to improve the quality of the article. All editing style comments were included in the revised manuscript. We also revised the images according to your suggestions. We have compiled your comments and our responses below:
Point 1- Line 249: “It should also be mentioned that the obtained results differ from the phase transformation calculations (Figure 3), “ - Please, explain why?
Response 1: The accuracy of the phase transformation simulations depends, of course, on the assumed model and material data/base. Commercial software like a JMatPro or ThermoCalc enables the proper estimation of the kinetics of phase transformations of the majority of materials. However, it is difficult to estimate the specific conditions related to possible variables (e.g. homogeneity of the chemical composition of austenite, initial microstructure, heating and cooling conditions, specific austenitization conditions, prior austenite grain size distribution, reverse transformation kinetics, and more). For this reason, the developed experimental results reflect the kinetics of phase transformations of the designed steel for specific austenitization conditions. The difference in the obtained results depends directly on the implemented variables and models for calculations, which in comparison to dilatometry, does not cover all features.
Point 2- Line 282: “It should also be highlighted that no phase transformation occurred in the range of 450-550 °C (Figure 8c). “ - that is weird. Specify the longer holding time.
Response 2: The maximum time of isothermal holding was adjusted to 68 h, which is a relatively long isothermal holding time. In our opinion, due to the purpose of this research, longer isothermal holding is not justified.
Point 3- Caption of Figure 8: please, specify the holding time for each temperature
Response 3: We specified a holding time for each temperature in the figure’s caption
“Figure 8. Microstructures of experimental steel after isothermal heat treatment at different temperatures. (a) 750 °C for 20 h; pearlite. (b) 575 °C for 68 h; pearlite, degenerated pearlite, partially spheroidal. (c) 450 °C for 68 h; lack of identified transformation, martensite. (d) 400 °C for 64 h; partially transformed upper bainite and marten-site. (e) 350 °C for 3 h; lath morphology of bainitic ferrite and retained austenite (blocks and films). (f) 275 °C for 15 h; lath morphology of bainitic ferrite, retained austenite (blocks and films). Scanning electron microscopy, etched by nital”
Point 4- Line 302. “Therefore, it was stated that there is no transformation in this temperature range, which is consistent with other studies[45]. “ Please provide a reason with fundament
Response 4: According to your comment we provided the explanation of this phenomenon:
Line 308. “It may be explained by the solute drag like effect (SDLE) caused by alloying additives (Cr, Mo, Ni) that significantly hinder the diffusion of carbon[42]. As a result, the in-complete transformation phenomenon or even a lack of transformation occurring between the ferrite/pearlite and the bainite transformation fields may be observed[43].”
Point 5- Table 4. Add a column specifying the transformation product.
Response 5: The product of bainitic transformation in all temperatures (275-375°C) was bainitic ferrite, retained austenite, and a limited fraction of carbide precipitates, which was also described in the further part of the manuscript. A detailed microstructural analysis was carried out, therefore, to improve the readability of this table, we decided to preserve it in its current form.
If the answers are not exhaustive, we will gladly answer the next questions.
Thank you for your consideration
Sincerely
Aleksandra Królicka
Reviewer 2 Report
1. Line 243: “bainitic structure was lower than 2 °C/s and lower than 0.5 °C/s.” lower than 0.5 °C/s?
2. Line 297: “the minimum time to form bainite time (tBs)”, here the last “time” is unnecessary.
Author Response
Dear Reviewer,
Thank you for your review of our manuscript and a few important comments. We hope that with their help we were able to improve the quality of the article. We have compiled your comments and our responses below:
Point 1- Line 243: “bainitic structure was lower than 2 °C/s and lower than 0.5 °C/s.” lower than 0.5 °C/s?
Response 1: We revised mentioned sentence:
Line 243. “bainitic structure was lower than 2 °C/s and higher than 0.5 °C/s.”
Point. 2. Line 297: “the minimum time to form bainite time (tBs)”, here the last “time” is unnecessary.
Response 2: Thank you for your comment. We revised the sentence according to your suggestion.
Line 301. “the minimum time to form bainite (tBs)”
If the answers are not exhaustive, we will gladly answer the next questions.
Thank you for your consideration
Sincerely
Aleksandra Królicka
Reviewer 3 Report
The article reports a novel bainitic steel with increased thermal stability of the structure at elevated temperatures. This method provides a new and interesting strategy to enhance the thermal stability of bainitic structure, which merits publication. However, there are several points that need to be addressed by the authors before publication is possible - these are listed below:
1. The authors suggest that the new steel has improved in-use properties in the abstract. However, there are no experimental results involving the in-use properties.
2. How about the mechanical properties of the new steel, such as tensile strength, yield strength and elongation. Are they improved or decreased compared with nanocrystalline bainitic steels? Can it substitute for the nanocrystalline bainitic steels?
Author Response
Dear Reviewer,
Thank you for your review of our manuscript and a few important comments. We have compiled your comments and our responses below:
Point 1- The authors suggest that the new steel has improved in-use properties in the abstract. However, there are no experimental results involving the in-use properties.
Response 1: Thank you for your comment. We agree that we have used a general concept of in-use properties in the abstract. In our previous work (doi: 10.1016/j.matdes.2021.110143), we related selected in-use properties (weldability, operating at elevated temperatures, and corrosion resistance) to the thermal stability of the structure. Therefore, our principal goal was to enhance thermal stability with simultaneously controlling the bainite transformation kinetics and structure morphology. We have clarified the abstract and the introduction to indicate how we expressed the concepts of improving the in-use properties of the designed steel.
Line 13: The material was characterized by improved in-use properties, expressed as the thermal stability of the structure, compared to nanocrystalline bainitic steels with a limited fraction of carbide precipitations.
Line 34. “The concept of enhancing in-use properties by improving the thermal stability of bainitic steels has been extensively described in previous work [7]. “
Point. 2. How about the mechanical properties of the new steel, such as tensile strength, yield strength and elongation. Are they improved or decreased compared with nanocrystalline bainitic steels? Can it substitute for the nanocrystalline bainitic steels?
Response 2: We fully agree with the above comment and the significance of mechanical properties testing. These tests will be carried out in the near future, especially considering the elevated operating temperature. In this work, we focused on the design process and the characterization of the new material in terms of phase transformations. Our intention was to present a complete characterization of this material, including experimental CCT and TTT diagrams, and to focus on the kinetics of the bainite transformation. It is known that thermal stability depends on the morphology of the matrix, and for this reason, we focused on the influence of the bainitic transformation temperature on the refinement level of bainitic ferrite and blocky retained austenite. We also evaluated the thermal stability during the tempering process as a validation of our design process. In addition, the scope of these studies may also be interesting for potential readers due to the included material data, which are also a source for designing the heat treatment of other medium carbon nanocrystalline bainitic steels.
If the answers are not exhaustive, we will gladly answer the next questions.
Thank you for your consideration
Sincerely
Aleksandra Królicka